# Prognostic Factors Analysis of Metastatic Recurrence in Cervical Carcinoma Patients Treated with Definitive Radiotherapy: A Retrospective Study Using Mixture Cure Model

**DOI:** 10.3390/cancers15112913

**Published:** 2023-05-25

**Authors:** Xiaxian Ou, Jing You, Baosheng Liang, Xiaofan Li, Jiangjie Zhou, Fengyu Wen, Jingyuan Wang, Zhengkun Dong, Yibao Zhang

**Affiliations:** 1Department of Biostatistics, School of Public Health, Peking University, Beijing 100191, China; 1810306124@pku.edu.cn (X.O.); actionsafe@pku.edu.cn (J.Z.); wjy123@bjmu.edu.cn (J.W.); 2Key Laboratory of Carcinogenesis and Translational Research (Ministry of Education/Beijing), Department of Radiation Oncology, Peking University Cancer Hospital & Institute, Beijing 100142, China; youjing0619@163.com (J.Y.); lxflp@163.com (X.L.); dongzhengkun@stu.pku.edu.cn (Z.D.); zhangyibao@pku.edu.cn (Y.Z.); 3Institute of Medical Technology, Peking University Health Science Center, Beijing 100191, China; wenfy@stu.pku.edu.cn

**Keywords:** brachytherapy, cervical carcinoma, metastatic recurrence, mixture cure model, activity of radioactive source

## Abstract

**Simple Summary:**

Definitive chemoradiotherapy is the standard treatment for locally advanced cervical carcinoma. The risk of local recurrence is usually low for cervical carcinoma patients treated with radical radiotherapy (external beam radiotherapy and brachytherapy). However, some of the patients relapsed with metastatic recurrence. The purpose of this study was to investigate prognostic factors associated with metastatic recurrence in cervical carcinoma treated with definitive radiotherapy. In addition to stage, our study found that age and radioactive level of brachytherapy significantly affected cervical cancer. An important finding was the interaction between age and source activity. In specific terms, brachytherapy with a high activity of radioactive source significantly benefits cervical carcinoma patients no more than 53 years old by prolonging the metastatic recurrence-free survival time compared with that using a low activity of radioactive source. Brachytherapy with mild source activity is more suitable for elderly patients.

**Abstract:**

Objectives: This study aims to identify prognostic factors associated with metastatic recurrence-free survival of cervical carcinoma (CC) patients treated with radical radiotherapy and assess the cure probability of radical radiotherapy from metastatic recurrence. Methods: Data were from 446 cervical carcinoma patients with radical radiotherapy for an average follow up of 3.96 years. We applied a mixture cure model to investigate the association between metastatic recurrence and prognostic factors and the association between noncure probability and factors, respectively. A nonparametric test of cure probability under the framework of a mixture cure model was used to examine the significance of cure probability of the definitive radiotherapy treatment. Propensity-score-matched (PSM) pairs were generated to reduce bias in subgroup analysis. Results: Patients in advanced stages (*p* = 0.005) and those with worse treatment responses in the 3rd month (*p* = 0.004) had higher metastatic recurrence rates. Nonparametric tests of the cure probability showed that 3-year cure probability from metastatic recurrence was significantly larger than 0, and 5-year cure probability was significantly larger than 0.7 but no larger than 0.8. The empirical cure probability by mixture cure model was 79.2% (95% CI: 78.6–79.9%) for the entire study population, and the overall median metastatic recurrence time for uncured patients (patients susceptible to metastatic recurrence) was 1.60 (95% CI: 1.51–1.69) years. Locally advanced/advanced stage was a risk factor but non-significant against the cure probability (OR = 1.078, *p* = 0.088). The interaction of age and activity of radioactive source were statistically significant in the incidence model (OR = 0.839, *p* = 0.025). In subgroup analysis, compared with high activity of radioactive source (HARS), low activity of radioactive source (LARS) significantly contributed to a 16.1% higher cure probability for patients greater than 53 years old, while cure probability was 12.2% lower for the younger patients. Conclusions: There was statistically significant evidence in the data showing the existence of a large amount of patients cured by the definitive radiotherapy treatment. HARS is a protective factor against metastatic recurrence for uncured patients, and young patients tend to benefit more than the elderly from the HARS treatment.

## 1. Introduction

Cervical carcinoma (CC) is the fourth most frequently diagnosed cancer and the fourth leading cause of death in women, with an estimated 604,000 new cases and 342,000 deaths worldwide in 2020 [1]. The age-standardized incidence rate and mortality rate in China, in 2016, were 11.34 (1/10^5^) and 3.36 (1/10^5^), respectively [2]. Although the risk of local recurrence is relatively low, part of the patient group relapsed with metastatic recurrence [3], which accounts for more than 50% of all recurrence [4]. Metastatic recurrence will develop in approximately 15–35% of women with cervical cancer, usually within a few years of primary treatment [5,6]. Once the recurrence happens, the patient faces limited treatment options and a poor prognosis [7]. Thus, CC is still a burden to public health and it is crucial to identify patients with a high risk of metastatic recurrence as soon as possible.

There is well-documented evidence which bears a significant influence on prognoses, such as the FIGO stage [8,9] and concurrent chemotherapy [6,10]. However, some controversies remain to be solved; for instance, some studies reported inconsistent results for age [11,12] and differentiation [13,14]. Anyway, though a large number of studies have explored different biological effects between high dose rate(HDR) and low dose rate(LDR) in brachytherapy y [15,16], there has been limited study around the impact of higher or lower source activity on patients’ metastatic recurrence within the dose-rate scope of HDR modality (>12 Gy/h) [17]. In practice, the dose-rate effect is complex because it varies widely at any reference point during treatment and the HDR 192Ir has a short half-life (74 days). Large varieties of dose rate in treatment delivery due to delayed source replacement that caused by unexpected public events such as specific quarantine during COVID-19 pandemic, administrative regulations, or supply shortage is a great challenge in the evaluation of treatment effectiveness. Thus, it is also worth exploring the metastatic recurrence risk under great variation in the activity of radioactive source(ARS) [18].

In prognostic studies for CC patients treated with radical radiotherapy, the Cox proportional hazards (PH) model has often been used to identify the prognostic factors for recurrence [9,19]. The ordinary Cox PH model assumes that all patients will eventually experience relapse. However, in fact, modern definitive radiotherapy could provide effective improvements for CC patients. A study by Karlsson et al. reported a low metastatic recurrence rate of 16.1% in brachytherapy [6]. Another study showed that the five-year distant metastasis-free survival of CC patients with CT-guided brachytherapy was 80.0% [20]. The low rates of metastatic recurrence reported in previous studies motivated us to think that perhaps many metastatic recurrence-free patients over five years had already been cured, or nearly cured (or effectively cured), so that they would not experience metastatic recurrence at all and, hence, we could not observe the outcome in practice even over a long follow-up period such as five or eight years. However, most existing literature simply treats the observation of potentially cured patients as censored data without accounting for the effect of potentially cured patients.

The mixture cure model is specifically developed to handle the above challenges [21]. In the mixture cure model, a portion of the subjects are deemed immune from the outcome of interest; the outcome of these subjects will never be observed no matter how long the follow-up time is. In research of oncology, cure models are appealing because they make it possible to know if and when a survivor can be considered cured of his or her cancer [22]. The mixture cure model is particularly helpful in evaluating the survival curves of uncured patients and in accurately assessing the hazards ratio (HR) of prognostic factors when the outcomes are partially observed from unknown cured patients [23].

To the best of our knowledge, so far there is limited study considering prognostic factor analysis for cervical carcinoma patients using a mixture cure model [24]. This study aims to assess prognostic factors associated with MRFS of cervical carcinoma (CC) patients treated with radical radiotherapy and assess the cure probability of radical radiotherapy from metastatic recurrence.

## 2. Materials and Methods

### 2.1. Patients

This study enrolled 747 women with pathologically confirmed FIGO 2009 (International Federation of Gynecology Oncology 2009) stage IA-IVB CC, who were treated with external beam radiotherapy and intracavitary brachytherapy under consistent protocols at the Department of Radiation Oncology, Peking University Cancer Hospital & Institute, between April 2011 and April 2017 [17]. In this study, patients were excluded due to the following reasons: (1) treatment interruption by massive bleeding or uterine perforation; (2) intra-course source replacement; (3) patients with non-squamous cell carcinoma; (4) observations had insufficient and invalid medical records; (5) follow-up time was less than 1 month; (6) patients diagnosed with distant metastasis of cervical carcinoma before brachytherapy. The remaining 446 patients were included for analysis, as seen in Figure 1.

### 2.2. Brachytherapy

All patients received pelvic contract-enhanced MRI, abdominal contract-enhanced CT, thoracic CT, and lymph node ultrasound before treatment. A prescribed dose of 45 Gy was delivered in 25 daily fractions to the planning target volume using intensity-modulated radiotherapy (IMRT) or volumetric-modulated arc therapy (VMAT) methods. Patients received an intravitreal brachytherapy boost utilizing the Ir-192 approach in three to four weeks after the first fraction of the external beam radiation portion. The equivalent dose in 2 Gy (EQD2) (assuming an α/β ratio of 10) to point A (a reference location 2 cm superior and 2 cm lateral to the central cervical os) ranged from 80 Gy to 85 Gy. The clinical target volume (CTV) covered the gross disease, corpus, whole uterus, parametria, sufficient vaginal margin from the gross disease (at least 3 cm), presacral nodes, and nodal volumes at risk. A simultaneous integrated boost regimen of 60 Gy in 25 fractions was recommended for the affected lymph nodes for patients with gross lymphadenopathy. The external and brachytherapy treatment planning was performed on a Varian Eclipse system (Varian Medical System, Palo Alto, CA, USA) and an Oncentra Brachy system (Elekta AB, Stockholm, Sweden), respectively. Intracavitary brachytherapy was delivered on a microSelectron Digital (HDR-V3) Brachytherapy Afterloader (Elekta Inc., Atlanta, GA, USA) using an Ir-192 source.

To be more clinically relevant, the mean dose rate (MDR) was used for this study instead of the encapsulated source activity [25]. To simplify the issue, patients with source replacement during their HDR treatment courses were excluded from this study. The average MDR value represented the dose rate of each patient of the consecutive treatment fractions. To differentiate from the naming of HDR vs. LDR, high activity of radioactive source and low activity of radioactive source within the dose rate range of HDR were abbreviated as HARS and LARS respectively.

In compliance with the guidance of NCCN (National Comprehensive Cancer Network) Guidelines version 1.2020 for cervical cancer [26], the patient received treatment with or without concurrent chemotherapy. Some patients did not receive concurrent chemotherapy because of the following reasons: (1) patients were of an early stage; (2) patients were too old or too weak to receive chemotherapy; (3) patients refused chemotherapy; (4) patients’ white blood cell counts were too low during brachytherapy. Meanwhile, according to this guideline, it was attempted to limit the entire radiotherapy course including both external beam radiotherapy and brachytherapy to 56 days (8 weeks). The enrolled patients’ median length of therapy overall was 43 days (range: 20 to 75 days).

### 2.3. Follow-Up

The longest follow-up in our study was 8.49 years. Follow-ups were performed every three months for the first two years; every six months from the third to the fifth year; and every year beginning with the sixth year. Examinations comprised physical examination, blood routine, liver and kidney function, tumor marker, pelvic MRI (pelvic CT for those not suitable for MRI), abdominal CT, thoracic CT, and lymph node ultrasonography, etc. All radiographic imaging was evaluated and reconfirmed by at least two experienced radiologists. According to RECIST 1.1 (Response Evaluation Criteria in Solid Tumors), which is generally used for solid tumors to reflect the acute effect of treatment, the 3rd-month outcome was assessed based on patients’ responses to radical radiotherapy [27], including complete response (CR), partial response (PR), stable disease (SD), and progressive disease (PD), respectively. Metastatic recurrence is chosen as the long-term outcome. In this study, metastatic recurrence-free survival (MRFS) was defined from the start of radiotherapy to distant metastasis.

### 2.4. Statistical Analysis

Clinical category characteristics for patients were reported as numbers of cases and percentages, and differences between subgroups were compared with the Chi-square test and Fisher’s exact test. Continuous variables were reported as medians and interquartile ranges (25th to 75th percentiles), and differences between subgroups were compared using the Wilcoxon test. Metastatic recurrence-free survival curves were obtained using the Kaplan–Meier estimator and the difference was compared using a log-rank test. The significance of cure probability of the treatment was examined under the framework of a mixture cure model using a nonparametric hypothesis test developed by Laska et al. [28].

A mixture cure model was then applied to assess prognostic factors associated with MRFS of CC patients. This model consists of two parts: a logistic incidence model and a Cox PH latency model. The survival function of the mixture cure model can be written as:(1)$$S\left(t|x,z\right)=1-\pi \left(z\right)+\pi \left(z\right)S_{u}(t|x)$$
where π(*z*) is the probability of a patient being uncured (patients are susceptible to metastatic recurrence), it depends on *z* and is depicted by the logistic incidence model, and $S_{u}(t|x)$ is the metastatic recurrence-free survival probability of uncured patients depending on *x* (latency model). 

In subgroup analysis, propensity score matching (PSM) based on a logistic regression model was performed to balance baseline covariates (tumor differentiation, tumor stage, mean dose, treatment duration, and concurrent chemoradiation) between HARS and LARS.

All statistical analyses were implemented using R software (version 4.1.l). The mixture cure model was implemented using the mixcure package. PSM was performed using the MatchIt package. A *p*-value smaller than 0.05 was considered statistically significant in the analyses.

## 3. Results

### 3.1. Characteristics of Patients and Metastatic Recurrence

The baseline characteristics of the study population (446 patients) are summarized in Table 1. The median follow-up time was 3.82 years (range = 48 days to 8.49 years). The median age of the patients was 53 years (range = 25–83 years). A total of 81 patients (18.2%) and 46 patients (10.3%) developed metastatic and local recurrence, respectively. The 1-year, 3-year, and 5-year MRFS rates were 90.8%, 84.5%, and 80.4%, respectively (see Figure 2A). The results of nonparametric hypothesis tests in Figure 2B showed that 3-year cure probability was significantly larger than 0, and 5-year cure probability was significantly larger than 0.7 but not significant for 0.8. Figure 2C shows the frequency of patients who were observed to have metastatic recurrence in different years, where 50.6% recurrence was observed within 1 year and nearly 99% recurrence within 5 years. Patients in more advanced stages (*p* = 0.005) and those with worse treatment response in the 3rd month (*p* = 0.004) were more likely to experience metastatic recurrence; significant correlation between local and metastatic recurrence was identified (*p* < 0.001). Anyway, the curves of Kaplan–Meier estimators for subgroups stratified by stage, the treatment response in the 3rd month, and local recurrence, respectively, and the corresponding *p*-values of the log-rank test are presented in Figure 3A–C. The MRFS curves and metastatic recurrence rates for patients with different activity of radioactive source and age groups are displayed, respectively, in Figure 3D–F.

### 3.2. Risk Factor Analysis Using Mixture Cure Model

The results of the mixture cure model are summarized in Table 2. In the latency model, mean dose rate was a significant protective factor against metastatic recurrence (HR = 0.682, *p* = 0.023). In the incidence model, the interaction between LARS and older age was significantly in favor of curing CC (OR = 0.839, *p* = 0.025). The incidence model also showed that locally advanced/advanced stage was a mild significant risk factor against the cure probability (OR = 1.078, *p* = 0.088).

### 3.3. Evaluation of Cure Probability and Median MRFS Times

Table 3 shows the estimated cure probabilities and the estimated median MRFS times for uncured patients stratified by clinical features. The mixture cure model revealed that in the entire study population, the overall probability of being cured from metastatic recurrence using radical radiotherapy was 79.2% (95% CI: 78.6–79.9%), and the overall median time of metastatic recurrence for uncured patients was 1.60 (95% CI: 1.51–1.69) years. The patients diagnosed with cancer at a locally advanced/advanced stage had lower cure probabilities; additionally, the median MRFS times were shorter in the patients with the characteristics of older age, LARS of radiotherapy, worse 3rd-month prognosis, and no concurrent chemoradiation. 

Figure 4 compares the estimated survival curves of metastatic recurrence by the regular Cox model and the mixture cure model, respectively. As we can see, the MRFS curves of uncured patients were all high under the Cox model, which means that they had a low risk of metastatic recurrence. However, the curves by the mixture cure model showed that uncured patients tended to have a higher risk of metastatic recurrence, which is more consistent with the clinical experience.

### 3.4. Cure Probability of ARS in Subgroup Analysis

The interaction of ARS and age is identified by the multivariate mixture cure model in Table 2. To further examine the effect of ARS, subgroup analysis was performed by stratifying age according to the median age (53 years old). To balance confounders between HARS and LARS, 93 data pairs were generated after PSM in patients no more than 53 years old, and 83 data pairs in patients greater than 53 years old. Mixture cure models were constructed in each subgroup, respectively, and the results summarized in Table 4. In the incidence model, LARS was significantly in favor of curing CC (OR = 0.294, *p* = 0.011) for older patients, but it shows a significant association with metastatic recurrence in younger patients (OR = 2.082, *p* = 0.045). In terms of cure probability predicted by the model, for older patients, LARS brachytherapy (91.1%) contributed to a higher cure probability than HARS brachytherapy (75.0%) with 16.1% difference, but for younger patients, there was a higher possibility of being cured through HARS brachytherapy (84.5%) than LARS (72.3%) with 12.2% difference. These results were consistent with the observed percentage of metastatic recurrence in Figure 2F.

## 4. Discussion

According to univariate analysis, patients with locally advanced/advanced stage had a higher risk of undergoing metastatic recurrence. Locally advanced/advanced stage was also a mild risk factor for metastatic recurrence in the mixture cure model. Consistent with our study, a number of studies have confirmed that FIGO has a strong association with metastatic recurrence [7,8,9]. The FIGO 2018 staging system reflects patient prognosis, as Qin noted [29]. Considering that patients with more advanced stages had a wider range of cancer lesions, and a higher probability of peripheral invasion and lymph node metastasis, the likelihood of recurrence for these patients was relatively higher [30]. Apart from stage, there was a connection between the 3rd-month prognosis and metastatic recurrence as well. The 3rd-month outcome was a signal to remind clinicians and patients to take some measures to prevent the poor situation of metastatic recurrence. Therefore, patients should take regular examinations after brachytherapy. Meanwhile, there might be same potential factors influencing both local recurrence and metastatic recurrence [31], so the local recurrence was equally worth attention.

Nonparametric test of cure probability shows that the 5-year cure probability for CC patients treated with radical radiotherapy was between 0.7 and 0.8 with statistical significance, which hints to us to examine the risk factors associated with metastatic recurrence for uncured patients using the mixture cure model. The consistent cure probability of 0.792 was indeed displayed in the model. For uncured patients, the estimated median metastatic recurrence time by mixture cure model was 1.60 years, and the estimated metastatic recurrence risk curve for uncured patients dropped rapidly even though the curve for overall patients demonstrated an optimistic result after treatment. 

The difference in cure rates between LARS and HARS for different age groups was worth investigating. Li et al. discovered in 2020 that LARS was clinically non-inferior to HARS for preventing metastatic recurrence [17]. Chen et al. demonstrated no dose-rate effect of a 192Ir source in HDR brachytherapy for cervical cancer in terms of pelvic control [18]. However, in this study, we discovered that HARS was a significant protected factor of MRFS for uncured CC patients, and there was significant interaction effect between ARS and ages on cure probability. In particular, after controlling covariates between HARS and LARS using PSM in each age stratification, the cure probability remained significantly higher in younger patients treated with HARS than that with LARS, and the cure probability of elder patients treated with LARS was higher than that treated with HARS. It was reported that a high dose rate led to a decreasing proliferation rate and cancer cell survival [32]; meanwhile, the damage to DNA increased in line with increased dose rates [33]. The HDR brachytherapy with HARS can effectively kill cancer cells so patients could benefit from it. However, HARS comes with some side effects. Suzuki et al. found that after experiencing HDR Ir192 brachytherapy, patients in the group with 2.4 cGy·m^−2^·h^−1^ or greater had a significantly greater frequency of late rectal bleeding than the other groups [34]. HDR radiation resulted in more lymphocyte depletion, one of the most radiosensitive cells which play an important role in keeping a stable immune system [35]. Therefore, we inferred that younger patients, who have a stronger ability to recover from normal tissue damage than elderly patients, could benefit more from HARS [36]. The elderly CC patients with compromised immune systems [37] and metabolic systems susceptible to being attacked [38] would be more likely to endure tremendous side effects, leading to normal cells suffering from radiotherapy and high metastatic recurrence rates. Although the mechanism is as yet unknown, it is clinically meaningful and could motivate further research. To better manage older and weak patients with cervical carcinoma, radiation oncologists should seek out training from geriatric specialists and work to understand the unique issues involved [39].

When it comes to the effect of chemoradiotherapy, a meta-analysis found that chemotherapy could lessen distant metastasis and prolong patient disease-free survival [40]. Zhang et al. reported that patients who received radiotherapy without chemotherapy were faced with a higher risk of metastatic recurrence (OR = 0.521, *p* = 0.040) [31]. In our study, although patients receiving concurrent chemotherapy had a lower rate of metastatic recurrence, and the coefficient showed its protective tendency in the mixture cure model, unfortunately, this effect was not significant in both univariate and multivariate analyses. Considering that most patients (86.1%) in our sample received concurrent chemotherapy, the statistical power was decreased. Moreover, different patients might receive various chemotherapy regimens, and this study only treated concurrent chemotherapy as a dichotomous variable. So there remains a challenge to further explore the effect of specific concurrent chemotherapy on metastatic recurrence.

This study has several limitations because of its retrospective design, where biases cannot be fully eliminated even when the covariates were controlled in the model. Due to inadequate or invalid medical records, a certain number of patients (16.2%) were excluded, which reduced the statistical power of the study and limited the promotion of our results to a wider population. Prospective studies are needed in the future to confirm associations between factors and metastatic recurrence more effectively. In addition, cervical cancer is very sensitive to radiotherapy [41], hence the low proportion of metastatic recurrence (18.2%) may require a larger population to obtain significant results instead of an insignificant trend in some studies. As for dose rate in HDR brachytherapy, our study did not consider the side effects of HARS; meanwhile, the optimal range of dose rate for each age group deserves further exploration in the future.

## 5. Conclusions

Over 70% of cervical cancer patients treated with radical radiotherapy were cured from metastatic recurrence. HARS is a protected factor against metastatic recurrence for uncured CC patients in radical radiotherapy. Younger patients tend to benefit more than elderly patients from HARS brachytherapy.

## Figures and Tables

**Figure 1 cancers-15-02913-f001:**
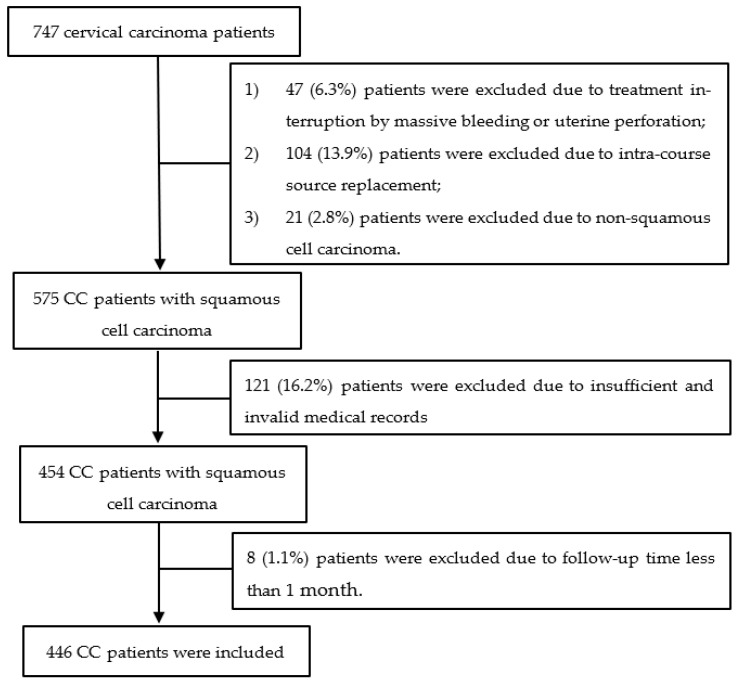
Sample selection profile.

**Figure 2 cancers-15-02913-f002:**
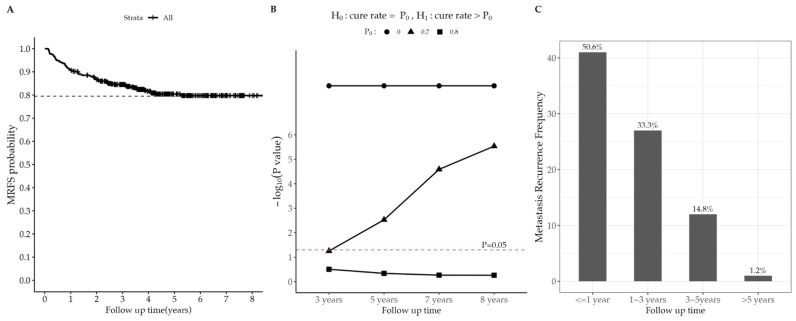
(**A**) Metastatic recurrence-free survival curves for the all patients; (**B**) *p*-value for the test of cure probability in different follow-up time; (**C**) Barplot of the metastatic recurrence frequency.

**Figure 3 cancers-15-02913-f003:**
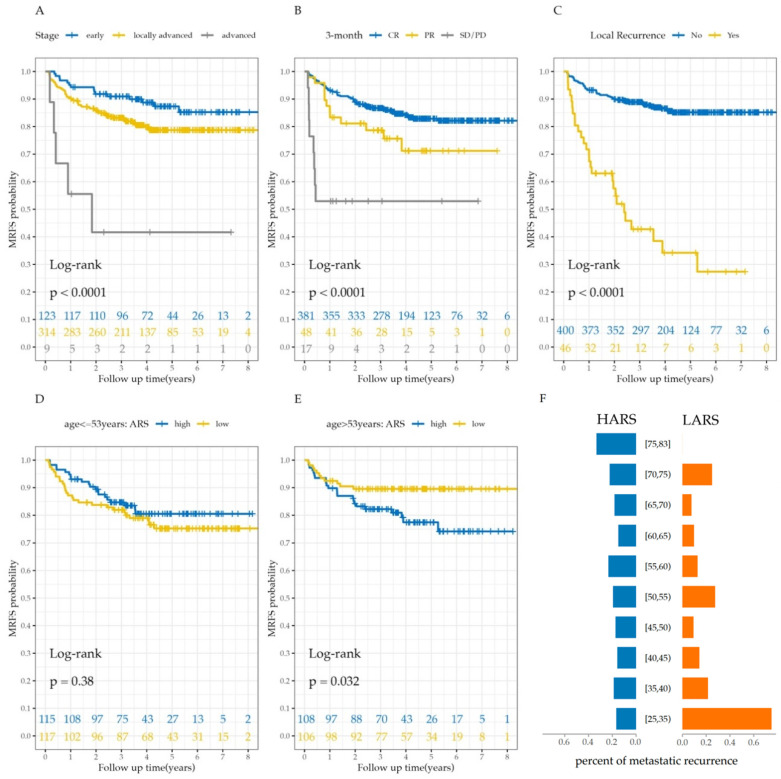
The K–M curves of metastatic recurrence in different subgroups: stage (**A**); treatment response in the 3rd month (**B**); local recurrence of cervical cancer (**C**); HARS and LARS for the patients with age no more than 53 (**D**); and for the group with age greater than 53 (**E**). The percentage of metastatic recurrence in different age groups (**F**).

**Figure 4 cancers-15-02913-f004:**
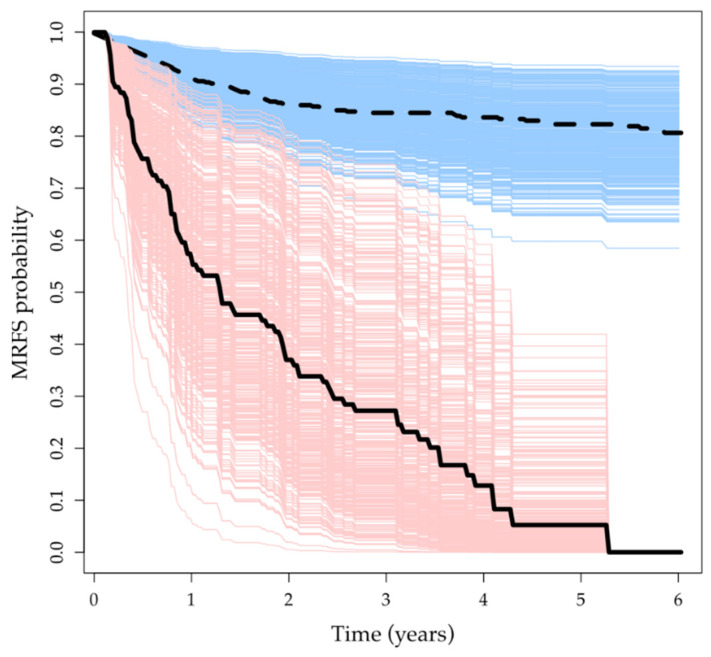
Estimated MRFS survival curves for each patient, where the blue lines were by regular Cox model, and the red lines were by mixture cure model. The solid black line and the dash black line indicate the empirical mean of all individualized MRFS survival curves by mixture cure model and regular Cox model, respectively.

**Table 1 cancers-15-02913-t001:** Baseline characteristics of all 446 patients, metastatic recurrence patients, and non-metastatic recurrence patients.

Variable	Overall	Non-Metastatic Recurrence	Metastatic Recurrence	*p*-Value
All patients	446	365	81	
Age (years)	53 (47, 59)	53 (47, 59)	52 (45, 57)	0.25
Differentiation [*n* (%)]				0.50
High	277	224 (61.4%)	53 (65.4%)	
Low	169	141 (38.6%)	28 (34.6%)	
Stage [*n* (%)]				0.005 *
Early	123	108 (29.6%)	15 (18.5%)	
Locally advanced	314	253 (69.3%)	61 (75.3%)	
Advanced	9	4 (1.1%)	5 (6.2%)	
Mean dose (cGy) [50th (25th, 75th)]	700 (620, 700)	700 (620, 700)	700 (620, 700)	0.26
Mean dose rate (cGy/h) [50th (25th, 75th)]	22120 (17,969, 30,085)	21908 (17,971, 30,026)	22,655 (17,788, 30,104)	0.89
Duration (days)	43.0 (41.0, 47.0)	43.0 (41.0, 46.0)	44.0 (41.0, 48.0)	0.068
Concurrent chemoradiation [*n* (%)]				0.33
No	62	48 (13.2%)	14 (17.3%)	
Yes	384	317 (86.8%)	67 (82.7%)	
3rd month [*n* (%)]				0.004 *
CR	381	320 (87.7%)	61 (75.3%)	
PR	48	36 (9.9%)	12 (14.8%)	
PD/SD	17	9 (2.5%)	8 (9.9%)	
Local recurrence [*n* (%)]				<0.001
No	400	346 (94.8%)	54 (66.7%)	
Yes	46	19 (5.2%)	27 (33.3%)	

NOTE: Continuous variables are reported as medians and interquartile ranges (25th to 75th percentiles); ‘*’ indicates Fisher exact test.

**Table 2 cancers-15-02913-t002:** Results of regression analysis using mixture cure model.

**Latency Model for Uncured Fraction**	**HR**	**SD**	***p*-Value**
Age	1.017	0.016	0.295
Stage	2.472	1.436	0.119
Mean dose	1.169	0.172	0.291
Mean dose rate	0.682	0.115	0.023
Concurrent Chemoradiation	0.714	0.233	0.304
**Incidence Model**	**OR**	**SD**	***p*-Value**
(Intercept)	1.157	0.112	0.132
Stage	1.078	0.047	0.088
Mean dose	1.028	0.020	0.146
Concurrent Chemoradiation	0.931	0.064	0.307
Age > 53	1.036	0.071	0.613
LARS	1.035	0.057	0.537
(Age > 53) × LARS	0.839	0.066	0.025

NOTE: Due to the low number of patients in advanced stage, stage was categorized into early stage (reference group) and locally advanced/advanced stage in the model. LARS and HARS have been categorized by the median mean dose rate.

**Table 3 cancers-15-02913-t003:** The estimated cure probabilities and median survival time of uncured patients stratified by clinical features.

Variable	*n*	Estimated Cure Probability × 100% (95% CI)	Median Time of Metastatic Recurrence in Uncured Patients, Years (95% CI)
All patients	446	79.2 (78.6–79.9)	1.60 (1.51–1.69)
Age			
≤53	232	77.0 (76.4–77.6)	1.69 (1.57–1.81)
>53	214	81.7 (80.5–82.9)	1.51 (1.38–1.64)
Differentiation			
High	277	79.0 (78.1–79.9)	1.54 (1.43–1.65)
Low	169	79.7 (78.6–80.8)	1.70 (1.56–1.85)
Stage			
Early stage	123	84.4 (83.1–85.6)	2.67 (2.50–2.84)
Locally advanced/ advanced	323	77.3 (76.6–78.0)	1.20 (1.14–1.25)
Mean dose rate			
High	223	76.4 (75.7–77.1)	2.03 (1.90–2.16)
Low	223	82.1 (81.0–83.2)	1.17 (1.09–1.26)
Concurrent chemo.			
No	62	75.8 (73.8–77.8)	1.22 (1.01–1.43)
Yes	384	79.8 (79.1–80.5)	1.66 (1.57–1.76)
3rd month			
CR	381	79.6 (78.9–80.4)	1.61 (1.52–1.71)
PR	48	77.0 (75.2–78.8)	1.64 (1.37–1.92)
PD/SD	17	77.4 (74.0–80.7)	1.31 (0.92–1.69)
Local recurrence			
No	400	79.5 (78.8–80.2)	1.60 (1.51–1.69)
Yes	46	77.1 (75.4–78.7)	1.62 (1.34–1.90)

NOTE: Continuous variables have been categorized on the basis of their median values.

**Table 4 cancers-15-02913-t004:** Subgroup analysis with PSM.

(a) age ≤ 53 years old, 93 pairs			
**Latency model for uncured fraction**	**HR**	**SD**	** *p* ** **-value**
Mean dose rate	0.786	0.218	0.386
**Incidence model**	**OR**	**SD**	** *p* ** **-value**
(Intercept)	0.184	0.055	0.000
LARS	2.082	0.761	0.045
(b) age > 53 years old, 83 pairs			
**Latency model for uncured fraction**	**HR**	**SD**	** *p* ** **-value**
Mean dose rate	0.847	0.223	0.529
**Incidence model**	**OR**	**SD**	** *p* ** **-value**
(Intercept)	0.334	0.094	0.000
LARS	0.294	0.142	0.011

## Data Availability

Research data are available upon reasonable request to the principal investigators under the commitment to following the hospital’s policy.

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
