# Peer review of "Prognostic Factors Analysis of Metastatic Recurrence in Cervical Carcinoma Patients Treated with Definitive Radiotherapy: A Retrospective Study Using Mixture Cure Model"

_cancers, 2023, doi:10.3390/cancers15112913_

Round 1

Reviewer 1 Report

The received manuscript is a statistical evaluation in treatment of Cervical Cancer as a moderately common disease.

The used techniques for treatment are not so novel especially in Brachytherapy part.

As you know these days according to EMBRACE I and II taking an MRI in the last week of EBRT and make decision about the technique of BT is common.

In your research there is no sign of intrestitial BT, which you know is a very useful device for local control, also preventing for mets.

As you mentioned upper stages have more complications and more metastasis probability. Your results emphasize these points.

At all I have some serious questions:

First of all in table 1 you reported the total dose equal to 700 Gy! What dose it mean?

I am not sure you can conclude from the gained results, that LDR is better than HDR for elder patients? There are not any other factors can effect on the results?

I think you have to re-write your manuscript to mention the results more accurate and also conclude. 

Author Response

Please see the attached response letter, thank you.

Reviewer 2 Report

 An issue that must be fixed by the authors is the description of the pre-treatment checkup (PET scanner? CT scan?).

Advanced stage must be better defined (the authors speaks about locally advanced versus advanced.) If advanced are metastatic diseases at the diagnosis, the treatment should not be definitive radiotherapy and should not be included in the analyses

The better outcome with LARS in  young patient must be used carefully; it is retrospective data, with numerous statistical manipulation, that rise the risk of bias. Why do the authors choose 53 yo as cut of age?

Author Response

Please see the attached response letter, thanks.

Round 2

Reviewer 2 Report

Corrections are ok, it is a weakness that the patient did not receive a PET scanner at the diagnosis but as it is a retrospective study, there is no way to modify it.